# Carbene-catalyzed double esterification enables enantioselective conformational self-locking of pillar[5]arenes

Vojtěch Dočekal [1] ✉, Ondřej Hladík [1], Ladislav Lóška [1], Martin Kamlar[1], Michael Franc[1], Ivana Císařová [2] & Jan Veselý [1] ✉

For their chirality, pillar[5]arenes stand out among the most versatile and widely utilized classes of organic host molecules, with numerous applications in materials science and biomedicine. The chirality of these macrocyclic compounds arises from conformational locking induced by sterically demanding substituents. However, preparing enantioenriched pillar[5]arenes often requires laborious chiral or diastereomeric separations. Here, we describe a simple and efficient metal-free protocol for oxidative double esterification of diformylpillar[5]arenes. This method provides a versatile and operationally straightforward route to highly enantioenriched products with broad functional group tolerance, as shown by incorporating biologically active and naturally derived fragments. Moreover, the reaction is readily scalable and enables subsequent derivatization, yielding rotaxane. Therefore, our organocatalytic method for pillar[5]arene self-locking not only streamlines the asymmetric synthesis of enantioenriched macrocyclic hosts but also opens new avenues for the design of chiral functional materials and host-guest systems.

*N*-heterocyclic carbenes (NHCs) have long been used as ligands for transition-metal complexes[1–3]. But since Rovis[4] and Bode[5] introduced rigid, electronically tunable and bench-stable chiral carbene precursors, NHC catalysis has become one of the most dynamic and versatile domains of organocatalysis[6]. Case in point, NHC-catalyzed oxidative esterification[7] proceeds through a Breslow intermediate subsequently oxidized to an activated acyl-azolium species that undergoes esterification with alcohols, yielding esters. Thanks to its operational simplicity and broad applicability, oxidative esterification has produced a wide range of complex (macrocyclic) chiral architectures[8], including planar chiral ferrocenes[9], paracyclophanes[10], inherently chiral calixarenes[11], and saddle-shaped lactones (Fig. 1A)[12–14]. Yet, in NHC catalysis, and even organocatalysis overall, chiral pillararenes have long been overlooked despite the importance of these inherently (or planar) chiral macrocyclic compounds, especially pillar[5]arenes.

First reported by Ogoshi in 2008[15], pillar[5]arenes have since demonstrated high potential as host molecules, encapsulating guest species within their cavity[16–18]. Their unique properties have led to numerous applications in host-guest chemistry[19], supramolecular assembly[20,21], materials science[22] and biomedicine[23,24]. Composed of five repeating *para*-phenylene-methylene units, pillar[5]arenes theoretically have eight conformers that interconvert via annular rotation, giving rise to inherently chiral stereoisomers[25]. But conformer interconversion can be effectively restricted (or locked) by introducing sterically demanding substituents. A study assessing the effect of hydroquinone alkyl-chain length on rotational barriers[26] has enabled the first isolation of configurationally stable enantiomers by introducing ten cyclohexylmethyl groups at both rims[27]. This approach was independently extended by Ogoshi[28] and Stoddart[29], who introduced sterically hindered (hetero)aryl substituents via palladium-catalyzed Suzuki-Miyaura coupling on *para*-phenylene units, yielding chiral HPLC-separable enantiomers. More recently, chiral auxiliary-based methods have been developed to separate diastereomeric mixtures[30–32]. However, the efficiency of such strategies is intrinsically

[1]Department of Organic Chemistry, Faculty of Science, Charles University, Hlavova 2030/8, Prague 2, Czech Republic. [2]Department of Inorganic Chemistry, Faculty of Science, Charles University, Hlavova 2030/8, Prague 2, Czech Republic. ✉e-mail: vojtech.docekal@natur.cuni.cz; jan.vesely@natur.cuni.cz

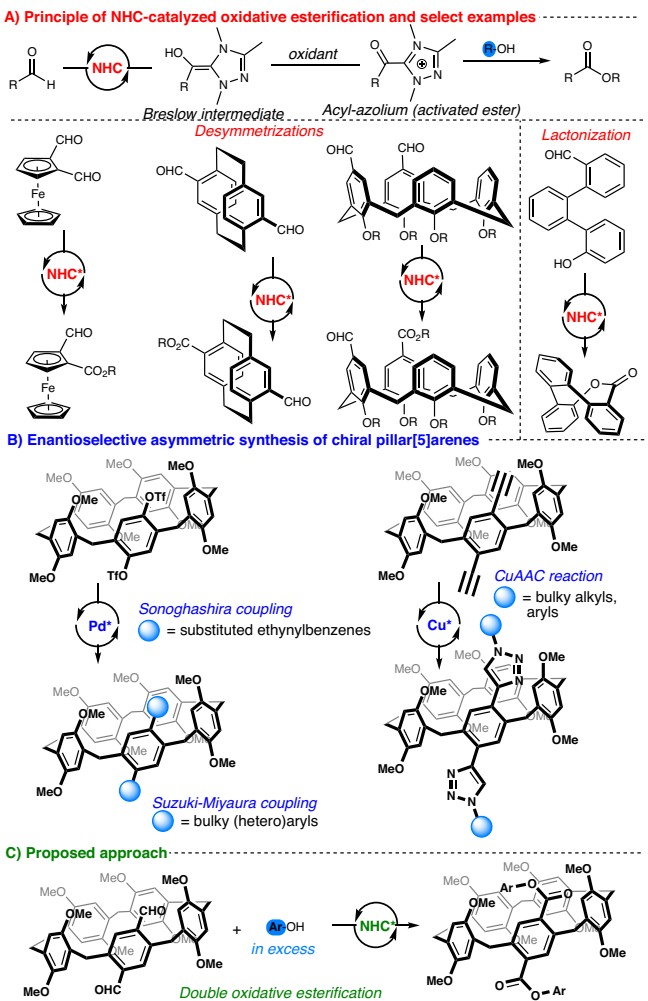

**A) Principle of NHC-catalyzed oxidative esterification and select examples**

Breslow intermediate    Acyl-azolium (activated ester)

*Desymmetrizations*          *Lactonization*

**B) Enantioselective asymmetric synthesis of chiral pillar[5]arenes**

*Sonoghashira coupling*
● = substituted ethynylbenzenes

*CuAAC reaction*
● = bulky alkyls, aryls

*Suzuki-Miyaura coupling*
● = bulky (hetero)aryls

**C) Proposed approach**

+ Ar-OH *in excess*

*Double oxidative esterification*

**Fig. 1 | Overview of NHC-catalyzed oxidative esterification and strategies for accessing chiral pillar[5]arenes. A** Principle of NHC-catalyzed oxidative esterification and select examples (highlighted in red). **B** Known asymmetric routes to chiral pillar[5]arenes (highlighted in blue). **C** Proposed approach (highlighted in green).

limited, only reaching 50% yield of the desired enantiomer and requiring stoichiometric amounts of enantiopure reagents. Nevertheless, continuous research efforts to synthesize inherently chiral pillar[5]arenes[33,34] have spurred the development of asymmetric catalytic methods. The latest advances have focused on enantioselective metal-catalyzed coupling reactions[35], including asymmetric Sonogashira[36] and Suzuki-Miyaura couplings[37–40], as well as an enantioselective copper-catalyzed azide-alkyne cycloaddition, affording inherently chiral pillar[5]arenes (Fig. 1B)[41].

Drawing on these advances in NHC-catalyzed oxidative esterification, this study aims to develop a metal-free and efficient enantioselective catalytic method for preparing chiral pillar[5]arenes. As an efficient route to enantioenriched products, we envisioned a double esterification of diformylpillar[5]arene (Fig. 1C).

## Results

### Optimization of the model reaction

At the outset of our study, we established and optimized a versatile route to diformylpillar[5]arene **1a** via DIBAL-H reduction of a previously reported dicyano derivative[42]. The detailed procedure for preparing starting materials **1** from commercially available 1,4-dialkoxybenzenes is provided in Supplementary Information (SI). Simply mixing the diformyl derivative **1a** with an excess of 2-naphthol (2.2

equiv), an achiral NHC precursor (*pre*-**C1**), an oxidant (Kharasch reagent, 3,3′,5,5′-tetra-*tert*-butyldiphenoquinone, DQ) and a base (cesium carbonate) afforded the expected diester *rac*-**3a** in 68% yield (Table 1, entry 1). By chiral HPLC, *rac*-**3a** was resolved, indicating configurational stability and successful conformation locking. Based on the results from this proof-of-concept experiment, we replaced the achiral NHC precursor with a chiral Bode catalyst (*pre*-**C2**, entry 2), which afforded the desired product **3a** with moderate enantiomeric purity (75:25 *er*). As expected[43], the enantiomeric ratio improved when using electron-withdrawing group-substituted Bode catalysts (*pre*-**C3,4**, entries 3,4). For instance, the reaction with the nitro-substituted Bode catalyst provided a high level of enantioselectivity (88:12 *er*, entry 4). The other catalysts, including *pre*-**C5**, failed to improve stereochemical outcomes. To enhance the efficiency and stereochemical performance of the model reaction, we systematically varied the solvent, oxidant, base, and other reaction parameters (for the full optimization, please refer to SI). The base strongly affected both yield and enantioselectivity. Although other (hydro)carbonates (entries 6,7) only slightly improved these parameters, potassium *tert*-butoxide and sodium acetate (entries 8,9) markedly enhanced the yield and enantioselectivity simultaneously. In particular, the reaction with sodium acetate afforded **3a** in high yield and excellent enantiomeric ratio (85%, 93:7 *er*). Further optimization highlighted dichloromethane as the most suitable solvent for this transformation. Notably, model reactions performed in ethyl acetate or tetrahydrofuran (entries 10, 11) resulted in nearly quantitative conversion to **3a**, albeit with slightly reduced enantiocontrol. Alternative oxidants proved ineffective, yielding only trace amounts of the desired product when using TEMPO (entry 12). Increasing the reaction temperature to 40 °C (entry 13) accelerated the reaction rate without compromising the yield or stereochemical outcome. Under these optimized conditions, we performed several control experiments (entries 14, 15). None of them led to further improvements. Consequently, the conditions established in entry 13 were adopted as optimal, affording **3a** in 82% yield (>90% yield for each esterification step) and 92:8 *er*.

### Reaction scope

After optimizing the reaction conditions, we investigated the scope of the conformation-locking esterification reaction using various rim-substituted diformylpillar[5]arenes and substituted aromatic alcohols (Fig. 2). The model reaction with the opposite enantiomeric form of the chiral catalyst (*ent*-*pre*-**C4**) provided matching yield and enantiomeric purity values (84%, 92:8 *er*) of the opposite enantiomeric product (*ent*-**3a**). Subsequently, we assessed the effect of alkyl substitution on the lower rim of the diformylpillar[5]arene (Fig. 2A). The esterification reaction of the ethoxy-derived starting material **1b** proceeded smoothly, giving rise to the desired product without any change in isolated yield and with only slightly reduced enantioselectivity (85:15 *er*). We postulated that the more rigid conformation of **1b** accounted for this decrease in enantioinduction. Further attempts at preparing a more sterically hindered starting material (R = Pr) failed (for details, refer to SI). Additionally, we extended the scope of various diformylpillar[5]arenes to include the α,β-unsaturated aldehyde **1c**. Due to decomposition of the starting material and product under the optimized conditions, the reaction rate and yield of **3c** (23%) decreased significantly, virtually abolishing enantioinduction. We postulate that the extended conjugated system shifts the reactive center away from the chiral environment, thereby preventing effective asymmetric induction.

We examined the scope and limitations of our method using various substituted naphthols (Fig. 2B). Broadly speaking, the reaction tolerated a wide range of naphthols bearing either electron-donating or -withdrawing groups at the 7- or 8-positions. In all cases, the expected products **3d-u** were prepared in high yields (36-93%) and with excellent enantioinduction (surpassing 90:10 *er*). For instance, the ester-derived product **3g** and the 7-methoxy derivative (**3m**) were

**Table 1 | Optimization of the conditions of the model reaction**

| Entry[a] | pre-Catalyst | Base | Time (h) | Yield[b] (3a, %) | er[c] (3a) |
|---|---|---|---|---|---|
| 1 | pre-**C1** | Cs$_2$CO$_3$ | 2 | 68 | 50:50 |
| 2 | pre-**C2** | Cs$_2$CO$_3$ | 2 | 69 | 75:25 |
| 3 | pre-**C3** | Cs$_2$CO$_3$ | 2 | 85 | 85:15 |
| 4 | pre-**C4** | Cs$_2$CO$_3$ | 3 | 59 | 88:12 |
| 5 | pre-**C5** | Cs$_2$CO$_3$ | 2 | 69 | 14:86 |
| 6 | pre-**C4** | Na$_2$CO$_3$ | 24 | 78 | 90:10 |
| 7 | pre-**C4** | NaHCO$_3$ | 48 | 86 | 91:9 |
| 8 | pre-**C4** | KOtBu | 2 | 43 | 93:7 |
| 9 | pre-**C4** | AcONa | 48 | 85 | 93:7 |
| 10[d] | pre-**C4** | AcONa | 72 | 98 | 90:10 |
| 11[e] | pre-**C4** | AcONa | 1 | 98 | 90:10 |
| 12[f] | pre-**C4** | AcONa | 72 | 12 | 90:10 |
| 13[g] | pre-**C4** | AcONa | 18 | 82 | 92:8 |
| 14[h] | pre-**C4** | AcONa | 24 | 28 | 91:9 |
| 15[e] | pre-**C4** | Cs$_2$CO$_3$ | 1 | 75 | 84:16 |

Er enantiomeric ratio.

[a]Reactions performed with **1a** (0.05 mmol), **2a** (0.11 mmol), base (0.10 mmol), DQ (0.11 mmol) and selected pre-catalyst (20 mol%) in DCM (1.0 mL) at room temperature (21–30 °C).
[b]Isolated yield assessed by column chromatography.
[c]Enantioselectivity determined by chiral HPLC analysis.
[d]EtOAc and
[e]THF used as solvent.
[f]TEMPO used as oxidant.
[g]Reaction performed at 40 °C.
[h]Reaction performed at 0 °C with THF as solvent.

isolated in high yields (93 and 87%) and with high enantiomeric purity (92:8 and 96:4 er, respectively). Tetrahydronaphthol esterification reached the highest enantiocontrol, affording product **3u** in 98:2 er.

Considering the above, we extended the substrate scope to phenols (Fig. 2C). In addition to phenol, both 4-methyl- and 4-bromophenol afforded the desired products **4a** and **4b** in high yields (74-90%). However, we were unable to resolve those products by chiral HPLC, and their specific rotations were zero. These results suggested ineffective conformational locking and the formation of achiral products. To gain deeper insights into the steric hindrance of those substituents, we assessed multidimensional steric parameters[44], namely Sterimol steric parameters[45]. Among them, the L parameter expresses the length of the substituent, which likely restricts the rotation of the ester through the pillar[5]arene annulus. As recently reported[46], methyl (L = 3.43 Å) and bromine (L = 4.29 Å) differ only slightly in length, thus lacking steric restriction. To validate this hypothesis, we selected a slightly more sterically demanding substituent that is, a tert-butyl group (L = 4.54 Å). The corresponding product **4d** was isolated in high yield (71%) and, moreover, in a configurationally stable and highly enantioenriched form (95:5 er). Subsequently, we employed 4-phenylphenol, which bears a much more spatially demanding substituent (L = 6.80 Å). This reaction afforded the expected product **4e** with a similar enantiomeric ratio of 93:7 er.

In addition, we investigated the scope of various aliphatic alcohols (Fig. 2D). The reaction with methanol afforded the product **4f** in high

isolated yield (71%). However, we were unable to resolve the enantiomers, indicating a lack of configurational stability. A similar stereochemical outcome was observed for all other aliphatic alcohols tested in this study. Significant product formation was observed only with primary alcohols; reactions with secondary or tertiary alcohols failed to yield the desired diesters. Furthermore, other than methanol, primary alcohols led to markedly lower reaction rates and yields. For instance, the reaction with 3-phenylpropanol afforded the desired product **4j** in only 26% yield. These results show that this method is more effective with less hindered primary alcohols.

With these results in hand, we assessed the effect of sterically demanding natural and biologically active phenols (Fig. 2E). All esterification reactions proceeded smoothly, affording the corresponding esters **4k-n** in good-to-high yields and with consistently high stereocontrol. For instance, the estrone-derived product **4l** was isolated in 93% yield and with a 10:1 diastereomeric ratio. To further probe chemoselectivity to aromatic alcohols, we compared the reactivity of a substrate containing both aromatic and aliphatic alcohols. The reaction with ezetimibe provided the expected product **4n** in a slightly higher yield (80 vs. 76%) than that of its secondary alcohol-protected analog **4m**, without byproducts. Conversely, the esterification of aliphatic alcohols, such as chemodeoxycholanol, proceeded with significantly lower reaction rates, without significant diester formation. Therefore, our organocatalytic method for conformation-locking esterification shows high chemoselectivity to

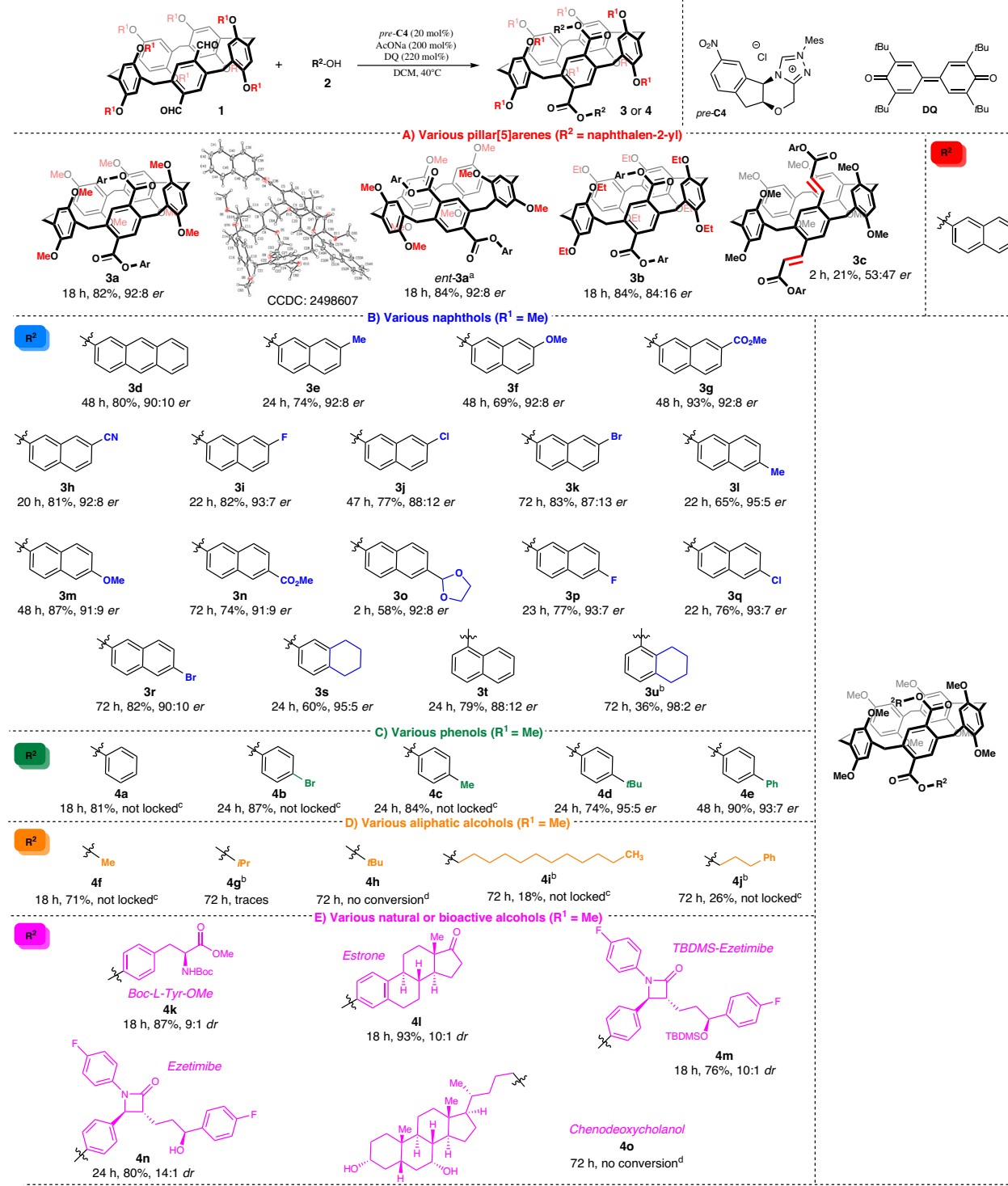

**Fig. 2 | Substrate scope leading to diester products. A** Various pillar[5]arenes (highlighted in red). **B** Various naphthols (highlighted in blue). **C** Various phenols (highlighted in green). **D** Various aliphatic alcohols (highlighted in orange). **E** Various natural or bioactive alcohols (highlighted in pink).

aromatic alcohols and compatibility with complex molecular structures.

To identify the structural features that govern this stereoinduction, we solved the structure of the products by X-ray crystallographic analysis. Because **3a** afforded racemic crystals (CCDC: 2498606), we analyzed the mother liquor, which contained virtually enantiopure **3a** (≥99.5:0.5 *er*). Recrystallization of the enantiopure sample produced single crystals suitable for the determination of the absolute

configuration (CCDC: 2498607). To further expand the scope of this method, we examined the esterification of unsymmetrical mono-aldehydes **1** (Fig. 3). Drawing from a previous report[38], we first tested the biphenyl-derived aldehyde **1c** (Fig. 3A). Under modified reaction conditions (1.2 equiv of alcohol and 1.2 equiv of DQ), we observed a slightly reduced reaction rate. Nevertheless, the expected unsymmetrical products were obtained in good yields (above 62%) and with high levels of stereocontrol. Subsequently, the monoesterified aldehyde

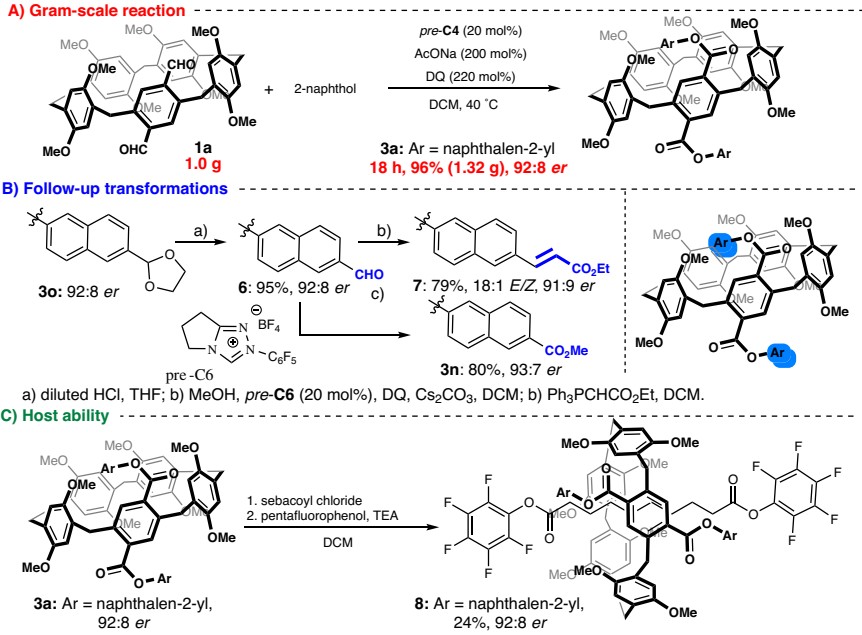

**Fig. 3 | Substrate scope leading to unsymmetrical products. A** Biphenyl-derived (highlighted in red). **B** Ester-derived (highlighted in blue).

**Fig. 4 | Gram-scale esterification and synthetic utility. A** Gram-scale reaction (highlighted in red). **B** Follow-up transformations (highlighted in blue). **C** Host ability (highlighted in green).

derived from **1a** under modified conditions over an achiral catalyst (please refer to SI for further details) was subjected to the optimized reaction. The expected product **3a** was isolated in a high yield (88%) and with a high enantiomeric ratio (93:7 *er*), in line with the diester-ification results (Fig. 2). So, enantioinduction occurs during the last esterification step, which is the second step of diesterification in the proposed mechanism (please refer to SI).

## Synthetic utilization of the chiral product

To assess whether the product was robust enough for further synthetic applications, we measured the configurational stability of **3a**. No significant decrease in enantiomeric ratio was observed at temperatures up to 70 °C, and the racemization barrier was 28.9 kcal·mol⁻¹ (for details, please refer to SI). In other words, the conformationally locked framework of **3a** prevents racemization, demonstrating its suitability

for further derivatization. To assess the practicality of our method, we performed a gram-scale reaction of **1a** under optimized conditions (Fig. 4A). The desired chiral product **3a** was isolated in a nearly quantitative yield (96%) and with high enantiomeric purity (92:8 *er*). In subsequent post-functionalization studies, we increased molecular complexity through transformations of the formyl group (Fig. 4B). Under acidic conditions, cleaving the dioxolane protecting group of **3o** afforded the corresponding dialdehyde **6** in a nearly quantitative yield. The dialdehyde **6** underwent a Wittig olefination, providing the expected alkene **7** in a high yield while retaining its stereochemical integrity. NHC-mediated oxidative esterification enabled post-functionalization of aldehydes, bypassing the conventional oxidation/esterification sequence, and providing the corresponding ester **3n** in a high yield. As mentioned above, pillar[5]arenes stand out for their potential as host molecules. To evaluate the host-guest properties of the enantioenriched diester (Fig. 4C), we followed previously reported conditions[47] using sebacoyl chloride and pentafluorophenol-derived stoppers in the presence of a host molecule **3a**. The desired rotaxane **8** was isolated in a moderate yield and with full retention of enantiopurity. The ability of **3** to retain its stability and enantiopurity during follow-up transformations showcases the value of pillar[5]arenes as chiral macrocyclic platforms for constructing interlocked architectures.

In summary, our straightforward organocatalytic method for (self)-locking conformational control of pillar[5]arenes provides versatile access to unique chiral macrocyclic products[48]. This operationally simple and highly efficient method shows excellent functional-group tolerance, enabling post-functionalization of natural and bioactive molecules. Moreover, the feasibility of gram-scale synthesis and synthetic utility underscores the broader significance of this approach. This strategy highlights the potential of diformylpillar[5]arene derivatives as valuable building blocks for asymmetric macrocyclic synthesis. Ongoing efforts in our laboratories to design inherently chiral molecules via organocatalytic transformations may lead to diverse applications.

## Methods
### Representative procedure
The vial (4 mL) was charged with **1** (0.05 mmol, 1.0 equiv.), the corresponding alcohol **2** (0.11 mmol, 2.2 equiv.), *pre*-**C4** (4.1 mg, 0.01 mmol, 0.2 equiv.), DQ (45.0 mg, 0.11 mmol, 2.2 equiv.) and AcONa (8.2 mg, 0.10 mmol, 2.0 equiv.) were dissolved in DCM (1.0 mL) at room temperature (~20 °C). The reaction mixture was heated to 40 °C (heating block), and the reaction mixture was stirred for the indicated time. Once the starting pillar[5]arene or the corresponding monoester was no longer detected by thin-layer chromatography (TLC), the reaction mixture was directly loaded into a silica gel column for chromatography, and the product was eluted with hexane/EtOAc mixtures.

## Data availability
The authors declare that the data supporting the findings of this study are available within the article and SI file. The primary NMR data generated in this study have been deposited in the Figshare repository under accession code (https://doi.org/10.6084/m9.figshare.30521552)[49]. The X-ray crystallographic coordinates for structures reported in this study have been deposited at the Cambridge Crystallographic Data Center (CCDC), under deposition numbers CCDC 2498606 (for *rac*−**3a**), 2498607 (for **3a**), and 2498608 (**3q**). These data can be obtained free of charge from The Cambridge Crystallographic Data Center via www.ccdc.cam.ac.uk/data_request/cif, or by emailing data_request@ccdc.cam.ac.uk, or by contacting The Cambridge Crystallographic Data Center, 12 Union Road, Cambridge CB2 1EZ, UK; fax: +44 1223 336033.

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

## Acknowledgements

The authors gratefully acknowledge the Czech Science Foundation (26-23610S, J.V.), Charles University Research Center program (UNCE/24/SCI/010, V.D.) and ERDF/ESF project TECHSCALE (CZ.02.01.01/00/22_008/0004587, M.F.) for financial support. The authors thank Dr. Carlos V. Melo (Charles University) for editing the manuscript. The authors gratefully acknowledge the staff of the Mass Spectrometry Group at IOCB Prague for conducting the mass spectrometric analyses of the samples.

## Author contributions

V.D. designed the project, conceived the study, and performed the synthesis. O.H., L.L., M.K., and M.F. performed the synthesis. I.C. performed X-ray analysis. J.V supervised the project. V.D. and J.V. wrote the manuscript. All authors have approved the final version of the manuscript.

## Competing interests

The authors declare no competing interests.
