## [Transparent Peer Review file · Nature Communications]

Carbene-Catalyzed Double Esterification Enables Enantioselective Conformational Self-Locking of Pillar[5]arenes

Corresponding Author: Professor Jan Vesely

Version 0:

Reviewer comments:

Reviewer #1

(Remarks to the Author)

Chiral macrocyclic compounds exhibit vast application in natural products, organic catalysis, and materials science, rendering the development of novel synthetic strategies for these compounds crucial. Asymmetric catalysis stands out as a direct and efficient method for synthesizing chiral molecules, and the asymmetric catalytic synthesis of chiral macrocycles has recently emerged as a research hotspot. Chiral pillar[5]arenes, as a new generation of supramolecular macrocyclic compounds, have found extensive applications in porous materials, molecular machines, and chiral luminescent materials due to their unique cavity structures and intriguing planar chirality. Currently, the synthesis of chiral pillar[5]arenes predominantly relies on chiral resolution, but the high preparation costs and limited separation scale restrict their practical application. Given the significant fundamental research value and broad application prospects of chiral pillar[5]arenes derivatives, there is an urgent need to develop novel asymmetric catalytic methods for the enantioselective synthesis of functionalized chiral pillar[5]arenes. In this manuscript, Vojtěch Dočekal, Jan Veselý and co-workers reported a simple and efficient NHC-catalytic oxidative double esterification protocol for the asymmetric synthesis of chiral pillar[5]arenes. This operationally simple and highly efficient method shows excellent functional-group tolerance, enabling post-functionalization of natural and bioactive molecules. Moreover, the feasibility of gram-scale synthesis and synthetic utility underscore the broader significance of this approach. Overall, this is an interesting work and I recommend publication of this paper on Nature Communications after the following revisions:

- 1) Based on the optimization table, we observed that entry 9 demonstrates superior results in terms of both yield and enantiomeric excess (ee) compared to entry 13. Why the authors selected entry 13 as the optimal condition for this model reaction?
- 2) In Figure 2c, were attempts made to extend the carbon chain or introduce bulkier tert-butyl groups to validate the hypothesis?
- 3) In the Supporting Information, the HPLC chromatograms corresponding to compounds 3l and 5a show weak signal intensity. To maintain data accuracy and reliability, it is essential to reacquire these spectras.
- 4) "ml" should be "mL" (everywhere).

Reviewer #2

(Remarks to the Author)

The manuscript by Veselý, Dočekal, and co-workers presents an oxidative NHC-catalyzed esterification strategy for the enantioselective synthesis of inherently chiral conformational self-locking Pillar[5]arenes. Historically, the synthesis of Pillar[5]arenes has primarily relied on metal-catalyzed coupling reactions. This study introduces an organocatalytic route for the synthesis of Pillar[5]arenes. Through extensive optimization of reaction conditions, the authors have identified ideal parameters for the efficient and highly enantioselective preparation of chiral Pillar[5]arenes. The Supplemental Information (SI) is well-structured, featuring clear spectra and comprehensive data characterization. Based on my assessment, I recommend the publication of this paper in Nature Communications, provided that the authors address the following issues: 1. The title "Organocatalytic Conformational Locking of Pillar[5]arenes" does not accurately reflect the research content. I suggest revising it to a more specific title that better encapsulates the essence of the study.

2. From the product 4h, it appears that the esterification reaction proceeds only with phenols and does not react with alcohols, even with the less sterically hindered primary alcohol, chemodexyholanol. Please provide an explanation for this observation. Additionally, would the reaction occur if methanol or ethanol were used? Would longer-chain phenylpropanols exhibit any enantioselectivity?
3. Under the optimized reaction conditions, what would be the expected outcome if the substituent R in substrate 1 were n-butyl or benzyl?
4. If the aldehyde group (-CHO) in substrate 1 were substituted with an unsaturated aldehyde, would the reaction still proceed under optimal conditions? Would this result in kinetic resolution (KR) or dynamic kinetic resolution (DKR)?
5. In the SI, the authors conducted a rotational barrier experiment for product 3a, indicating the temperature at which racemization occurs. For Pillar[5]arenes, could the authors derive a general conclusion regarding the stability of various substituted Pillar[5]arenes at room temperature through computational analysis or literature review?
6. The SI should include a detailed methodology for the growth of single crystals.
7. Some HPLC chromatograms in the SI exhibit irregular baselines, low absorbance peaks, or excessively long retention times, which may lead to inaccuracies in the reported enantiomeric excess (ee) values for products such as 3f, 3l, 3n, and 5b.
8. Please specify the source and/or preparation method for the pre-catalyst compounds C1-C27 in the SI. For new catalysts, complete characterization data should be provided. For known catalysts synthesized according to published methods, please include relevant references and submit one or more of the following for purity documentation: ¹H NMR or ¹³C NMR spectra.

Version 1:

Reviewer comments:

Reviewer #1

(Remarks to the Author)

The authors have effectively solved the problems, and the manuscript is recommended for publication.

Reviewer #2

(Remarks to the Author)

I have carefully reviewed the authors' response to my previous comments as well as the revised manuscript and Supplementary Information. I am pleased to note that the authors have addressed most of my concerns in a thorough and scholarly manner. The authors have gone to considerable lengths to address each point experimentally or with well-reasoned explanations. The new data enrich the study, clarify its scope and enhance the scholarly rigor of the manuscript. The revisions have substantially improved the work, and it now meets the high standards expected for publication in Nature Communications.

Therefore, I recommend acceptance of the revised manuscript in its current form.

Faculty of Science
CHARLES UNIVERSITY
IN PRAGUE

Department of
ORGANIC CHEMISTRY

Dr. Vojtěch Dočekal, Prof. Jan Veselý
Charles University, Faculty of Science,
Department of Organic Chemistry
Hlavova 2030, 128 43 Praha 2, Czech Republic
Phone: +420-221-951-305, fax: 420 221 951 326,
E-mail: vojtech.docekal@natur.cuni.cz, jan.vesely@natur.cuni.cz

Dear Reviewers,

Thank you for your detailed and thoughtful critiques. Enclosed, please find the revised manuscript (ID: NCOMMS-25-91878) originally entitled: "Organocatalytic Conformational Self-Locking of Pillar[5]arenes". We have addressed all comments and modified the manuscript and Supplementary Information accordingly. Below, please find our responses to your comments.

Reviewer #1 (Remarks to the Author):

Chiral macrocyclic compounds exhibit vast application in natural products, organic catalysis, and materials science, rendering the development of novel synthetic strategies for these compounds crucial. Asymmetric catalysis stands out as a direct and efficient method for synthesizing chiral molecules, and the asymmetric catalytic synthesis of chiral macrocycles has recently emerged as a research hotspot. Chiral pillar[5]arenes, as a new generation of supramolecular macrocyclic compounds, have found extensive applications in porous materials, molecular machines, and chiral luminescent materials due to their unique cavity structures and intriguing planar chirality. Currently, the synthesis of chiral pillar[5]arenes predominantly relies on chiral resolution, but the high preparation costs and limited separation scale restrict their practical application. Given the significant fundamental research value and broad application prospects of chiral pillar[5]arenes derivatives, there is an urgent need to develop novel asymmetric catalytic methods for the enantioselective synthesis of functionalized chiral pillar[5]arenes. In this manuscript, Vojtěch Dočekal, Jan Veselý and co-workers reported a simple and efficient NHC-catalytic oxidative double esterification protocol for the asymmetric synthesis of chiral pillar[5]arenes. This operationally simple and highly efficient method shows excellent functional-group tolerance, enabling post-functionalization of natural and bioactive molecules. Moreover, the feasibility of gram-scale synthesis and synthetic utility underscore the broader significance of this approach. Overall, this is an interesting work and I recommend publication of this paper on Nature Communications after the following revisions:

We sincerely thank the reviewer for such a thorough review of our manuscript and for the insightful suggestions.

1) Based on the optimization table, we observed that entry 9 demonstrates superior results in terms of both yield and enantiomeric excess (ee) compared to entry 13. Why the authors selected entry 13 as the optimal condition for this model reaction?

The reviewer asks a pertinent question. We selected a slightly elevated temperature (40 °C, entry 13) as the optimal condition for two main reasons. First, our preliminary results at room temperature revealed a kinetic bottleneck: while the conversion of the starting dialdehyde to the monoaldehyde intermediate was fast, subsequent conversion to the target diester was significantly slower. This incomplete conversion complicated purification; we struggled to separate the product from the residual monoester intermediate. Second, we chose 40 °C to ensure reproducibility. Performing the reaction slightly above room temperature avoided inconsistencies caused by fluctuations in laboratory temperature. Since we observed no significant erosion of optical purity at this temperature, we identified 40 °C as the most robust condition.

Faculty of Science
CHARLES UNIVERSITY
IN PRAGUE

Department of
ORGANIC CHEMISTRY

Dr. Vojtěch Dočekal, Prof. Jan Veselý
Charles University, Faculty of Science,
Department of Organic Chemistry
Hlavova 2030, 128 43 Praha 2, Czech Republic
Phone: +420-221-951-305, fax: 420 221 951 326,
E-mail: vojtech.docekal@natur.cuni.cz, jan.vesely@natur.cuni.cz

2) In Figure 2c, were attempts made to extend the carbon chain or introduce bulkier tert-butyl groups to validate the hypothesis?

We thank the reviewer for the insightful suggestion to strengthen our hypothesis regarding the effect of substituent bulkiness at the *para*-position of phenol. As suggested, we performed the esterification reaction with 4-*tert*-butylphenol. This resulted in the formation of the desired product (**4d** in the revised manuscript), which was prepared in high enantiomeric purity (95:5 *er*) and yield (71%) under optimal reaction conditions. This observation further validates our hypothesis regarding the predictability of efficient conformational locking based on the Sterimol *L* (length) parameter. More specifically, the *tert*-butyl group (calculated *L* = 4.54 Å) provided sufficient bulkiness to restrict rotation, in contrast to the bromine-substituted derivative (*L* = 4.29 Å). We believe that this impactful finding bolsters rational substituent design. Accordingly, we have incorporated these new results into the revised manuscript and Supplementary Information and, above all, we sincerely thank the reviewer for this suggestion.

3) In the Supporting Information, the HPLC chromatograms corresponding to compounds 3l and 5a show weak signal intensity. To maintain data accuracy and reliability, it is essential to reacquire these spectras.

We completely agree with the reviewer. Following through on our commitment to revising this section so as to enhance accuracy, we have re-evaluated all HPLC chromatograms and reintegrated the signals at the absorption maxima of the corresponding compounds. As a result, we have identified minor deviations in the calculated enantiomeric ratios and updated these values in the revised draft of our manuscript and in Supplementary Information.

4) "ml" should be "mL" (everywhere).

We have corrected this typo in the revised manuscript as well as in the Supplementary Information file.

Reviewer #2 (Remarks to the Author):

The manuscript by Veselý, Dočekal, and co-workers presents an oxidative NHC-catalyzed esterification strategy for the enantioselective synthesis of inherently chiral conformational self-locking Pillar[5]arenes. Historically, the synthesis of Pillar[5]arenes has primarily relied on metal-catalyzed coupling reactions. This study introduces an organocatalytic route for the synthesis of Pillar[5]arenes. Through extensive optimization of reaction conditions, the authors have identified ideal parameters for the efficient and highly enantioselective preparation of chiral Pillar[5]arenes. The Supplemental Information (SI) is well-structured, featuring clear spectra and comprehensive data characterization. Based on my assessment, I recommend the publication of this paper in Nature Communications, provided that the authors address the following issues:

We would like to thank the reviewer for carefully reading our manuscript and for making constructive suggestions to improve our study and for the positive feedback on the structure and content of our study.

1. The title "Organocatalytic Conformational Locking of Pillar[5]arenes" does not accurately reflect the research content. I suggest revising it to a more specific title that better encapsulates the essence of the study.

We understand the reviewer's concern about a potential mismatch between the title and the content of this article. To more clearly assess whether the title reflects the research content, we have carefully unpacked the title into its keywords as follows:

(i) **Organocatalytic** (ii) **Conformational Locking** of (iii) **Pillar[5]arenes**"

Faculty of Science
Charles University
Albertov 6, 128 43 Praha 2
www.natur.cuni.cz

Faculty of Science
CHARLES UNIVERSITY
IN PRAGUE

Department of
ORGANIC CHEMISTRY

Dr. Vojtěch Dočekal, Prof. Jan Veselý
Charles University, Faculty of Science,
Department of Organic Chemistry
Hlavova 2030, 128 43 Praha 2, Czech Republic
Phone: +420-221-951-305, fax: 420 221 951 326,
E-mail: vojtech.docekal@natur.cuni.cz, jan.vesely@natur.cuni.cz

First, the study does involve (i) NHC-mediated **organocatalysis**. Second, this approach is applied to (iii) **pillar[5]arenes**. Third, the aim is to induce (ii) **conformational locking**. In fact our title factually and topically aligns with the reviewer's overview of this study: "This study introduces an organocatalytic route for the synthesis of Pillar[5]arenes". Nevertheless, we accept that the issue may be rhetorical, not semantic. Therefore, we have edited the title to improve clarity and emphasis, underscoring the asymmetric organocatalytic strategy and its role in enabling conformational self-locking of pillar[5]arenes: "Carbene-Catalyzed Double Esterification Enables Enantioselective Conformational Self-Locking of Pillar[5]arenes".

We thank the reviewer for helping us boost the visibility of our study by improving its title.

2. From the product 4h, it appears that the esterification reaction proceeds only with phenols and does not react with alcohols, even with the less sterically hindered primary alcohol, chemodexycolanol. Please provide an explanation for this observation. Additionally, would the reaction occur if methanol or ethanol were used? Would longer-chain phenylpropanols exhibit any enantioselectivity?

Once again, we sincerely thank the reviewer for such a thorough assessment of our manuscript. Following the suggestion to investigate aliphatic alcohols for mechanistic clarification and scope interpretation, we first tested the reaction with methanol. With this nucleophile, the desired product (**4f**) was formed in high yield (71% after 18 hours); however, we were unable to resolve the enantiomers due to insufficient steric hindrance. We then tested secondary (propan-2-ol) and tertiary (*tert*-butanol) alcohols. With these alcohols, high steric demand inhibited the reaction; even after 72 hours, propan-2-ol yielded only the monoester, with traces of the desired product. Lastly, we assessed whether chain length induced conformational locking by testing lauryl alcohol and 3-phenylpropanol (an aliphatic alcohol with a phenyl stopper). Both substrates showed significantly lower reaction rates than methanol (e.g., 18% yield for lauryl alcohol) and failed to provide configurational stability. These results strengthen our understanding of the limitations of our method, so we have included these data in the revised draft of our manuscript and in Supplementary Information.

3. Under the optimized reaction conditions, what would be the expected outcome if the substituent R in substrate 1 were n-butyl or benzyl?

We thank the reviewer for this insightful question. Based on the organocatalytic esterification of the ethoxy-substituted dialdehyde (84% yield, 84:16 *er*), we postulate that the additional steric bulk interferes with the rotational freedom required for effective enantioinduction. According to the literature, more sterically hindered aliphatic substituents may effectively promote conformational locking (*J. Org. Chem.* **75**, 3268–3273 (2010), DOI: 10.1021/jo100273n). However, we faced severe synthetic hurdles when trying to validate the effect of a larger group (propoxy-). First, our standard cyanation approach to preparing the starting materials is highly sensitive to steric hindrance; while the methoxy-derivative proceeded quantitatively, the propoxy-derivative failed to convert into the corresponding dicyano pillar[5]arene (Figure 1).

Dr. Vojtěch Dočekal, Prof. Jan Veselý
Charles University, Faculty of Science,
Department of Organic Chemistry
Hlavova 2030, 128 43 Praha 2, Czech Republic
Phone: +420-221-951-305, fax: 420 221 951 326,
E-mail: vojtech.docekal@natur.cuni.cz, jan.vesely@natur.cuni.cz

Figure 1. General procedure for the preparation of starting diformyl derivatives.

Second, we tried an alternative strategy involving the global deprotection of **1a** followed by alkylation. Using boron tribromide yielded a tetrabromo derivative rather than the expected product (Figure 2). Furthermore, hydrolysis of this intermediate resulted in a complex, inseparable mixture (Figure 3). Consequently, we are confident that longer aliphatic derivatives are not accessible via these general methodologies. Considering these synthetic obstacles, we were unable to extend the scope to larger alkoxy derivatives. To document these limitations, we have added these detailed negative results to the revised Supplementary Information file.

Figure 2. Alternative synthetic route.

Dr. Vojtěch Dočekal, Prof. Jan Veselý
Charles University, Faculty of Science,
Department of Organic Chemistry
Hlavova 2030, 128 43 Praha 2, Czech Republic
Phone: +420-221-951-305, fax: 420 221 951 326,
E-mail: vojtech.docekal@natur.cuni.cz, jan.vesely@natur.cuni.cz

Figure 3. Stacked ^1H NMR spectra of tetrabromo derivative and crude mixture after hydrolysis.

4. If the aldehyde group (-CHO) in substrate 1 were substituted with an unsaturated aldehyde, would the reaction still proceed under optimal conditions? Would this result in kinetic resolution (KR) or dynamic kinetic resolution (DKR)?

We sincerely thank the reviewer for this constructive suggestion. We immediately undertook the synthesis of the suggested unsaturated dialdehyde. Initially, we attempted a stepwise modification of 1a. A Wittig reaction afforded the expected unsaturated ester in high yield (85%) as a single *E*-isomer. However, subsequent reduction with excess DIBAL-H yielded the allylic alcohol as an inseparable mixture with by-products. Oxidation of this crude intermediate using Dess-Martin periodinane also failed to provide a pure product. Consequently, we designed an alternative strategy using a direct Wittig olefination with a dioxolane-protected aldehyde reagent. We selected this approach to prevent over-reaction to dienals and to simplify purification. Meeting our expectations, this route proved effective: following the Wittig reaction and acidic deprotection, we isolated the desired unsaturated dialdehyde in high yield as a single *E*-isomer.

Figure 3. Procedures for preparing an unsaturated aldehyde

Faculty of Science
CHARLES UNIVERSITY
IN PRAGUE

Department of
ORGANIC CHEMISTRY

Dr. Vojtěch Dočekal, Prof. Jan Veselý
Charles University, Faculty of Science,
Department of Organic Chemistry
Hlavova 2030, 128 43 Praha 2, Czech Republic
Phone: +420-221-951-305, fax: 420 221 951 326,
E-mail: vojtech.docekal@natur.cuni.cz, jan.vesely@natur.cuni.cz

Prior to the organocatalytic esterification, we analyzed the starting dialdehyde by chiral HPLC. The absence of peak splitting confirmed that the material was achiral and suitable for the intended transformation. We subsequently subjected this dialdehyde to a reaction with 2-naphthol under optimized conditions. While the reaction was fast (2 hours), we observed significant decomposition of the starting material, resulting in a low yield (21%) of the desired product. Furthermore, the product was obtained in virtually racemic form. We postulate that the extended conjugated system shifts the reactive center away from the chiral environment of the catalyst (or alters the transition state geometry), thereby preventing effective asymmetric induction. Given the low yield and lack of enantioselectivity, we decided not to investigate dynamic kinetic resolutions with this substrate. However, the synthesis and reactivity of this novel diformyl derivative represent a valuable extension of the substrate scope. Therefore, we have included these results in the revised manuscript and Supplementary Information.

5. In the SI, the authors conducted a rotational barrier experiment for product 3a, indicating the temperature at which racemization occurs. For Pillar[5]arenes, could the authors derive a general conclusion regarding the stability of various substituted Pillar[5]arenes at room temperature through computational analysis or literature review?

We thank the reviewer for this insightful suggestion. Based on the experimentally determined racemization barrier ($\Delta G=28.97$ kcal/mol), we have determined that the half-life is approximately 3.0 years (1100 days) at room temperature using the Eyring equation. According to the classification of atropisomeric stability proposed by LaPlante and colleagues (*J. Med. Chem.* **54**, 7005–7022 (2011), DOI: 10.1021/jm200584g), this level of stability places our compounds firmly within Class 3 (stable atropisomers). This class is widely regarded as suitable for drug development, with numerous FDA-approved examples demonstrating similar rotational barriers. To further contextualize this stability, we compared our results to structurally related systems. Our derivatives showed significantly higher stability than pillararenes bearing linear alkoxy groups at both rims (*J. Org. Chem.* **75**, 3268–3273 (2010), DOI: 10.1021/jo100273n). This stability is comparable to that of disubstituted systems bearing diamidic motifs (*Symmetry* **11**, 773 (2019), DOI: 10.1021/jo100273n). Our barrier is slightly lower than some polyaromatic pillararenes formed via coupling reactions (highly dependent on aryl substitution, *Chin. Chem. Lett.* **36**, 111201 (2025), DOI: 10.1016/j.ccl.2025.111201) but remains well within the range required for configurational stability under standard conditions.

6. The SI should include a detailed methodology for the growth of single crystals.

We thank the reviewer for the suggestion to revise the methodological section of the SI to ensure the reproducibility of our findings. We now include detailed procedures for crystallization in the characterization data for compounds **3a** and **3q**.

7. Some HPLC chromatograms in the SI exhibit irregular baselines, low absorbance peaks, or excessively long retention times, which may lead to inaccuracies in the reported enantiomeric excess (ee) values for products such as 3f, 3l, 3n, and 5b.

Once again, we thank the reviewer for the insightful suggestion to revise this section for accuracy. We have re-evaluated all HPLC chromatograms and reintegrated the signals at the absorption maxima of the corresponding compounds. Unfortunately, we were unable to separate highly polar compounds with lowered retention times given the technical limit of our machine. These changes resulted in minor deviations in the calculated enantiomeric ratios. We have updated the corresponding values in the revised draft of our manuscript and in Supplementary Information.

Faculty of Science
Charles University
Albertov 6, 128 43 Praha 2
www.natur.cuni.cz

Faculty of Science
CHARLES UNIVERSITY
IN PRAGUE

Department of
ORGANIC CHEMISTRY

Dr. Vojtěch Dočekal, Prof. Jan Veselý
Charles University, Faculty of Science,
Department of Organic Chemistry
Hlavova 2030, 128 43 Praha 2, Czech Republic
Phone: +420-221-951-305, fax: 420 221 951 326,
E-mail: vojtech.docekal@natur.cuni.cz, jan.vesely@natur.cuni.cz

8. Please specify the source and/or preparation method for the pre-catalyst compounds C1-C27 in the SI. For new catalysts, complete characterization data should be provided. For known catalysts synthesized according to published methods, please include relevant references and submit one or more of the following for purity documentation: ^1H NMR or ^{13}C NMR spectra.

We sincerely thank the reviewer for such a thorough review and for ensuring the reproducibility of our work. In this study, we used both commercially available precursors and compounds synthesized according to well-established literature procedures. We confirmed that the characterization data for the synthesized precursors matched the reported values. For clarity, we added the corresponding section to the revised Supplementary Information file, explicitly listing the commercial sources or specific references to literature on all precursors used in this study.

Sincerely Yours

Vojtěch Dočekal and Jan Veselý